# Hybrid Nanofillers Creating the Stable PVDF Nanocomposite Films and Their Effect on the Friction and Mechanical Properties

**DOI:** 10.3390/polym14183831

**Published:** 2022-09-14

**Authors:** Karla Čech Barabaszová, Sylva Holešová, Lukáš Plesník, Zdeňka Kolská, Kamil Joszko, Bożena Gzik-Zroska

**Affiliations:** 1Nanotechnology Centre, CEET, VŠB-Technical University of Ostrava, 17. Listopadu 15/2172, 708 00 Ostrava-Poruba, Czech Republic; 2CENAB, Faculty of Science, J. E. Purkyně University, Pasteurova 15, 400 96 Ústí nad Labem, Czech Republic; 3Department of Biomechatronics, Silesian University of Technology, Roosevelta 40, 41-800 Zabrze, Poland; 4Department of Biomaterials and Medical Device Engineering, Faculty of Biomedical Engineering, Silesian University of Technology, 41-800 Zabrze, Poland

**Keywords:** PVDF nanocomposite films, zinc oxide, vermiculite, chlorhexidine, nanofillers, correlative imaging, surface roundness, Zeta-potential values, friction and mechanical properties

## Abstract

The solvent casting method was used for five types of polyvinylidene difluoride (PVDF) nanocomposite film preparation. The effect of nanofillers in PVDF nanocomposite films on the structural, phase, and friction and mechanical properties was examined and compared with that of the natural PVDF film. The surface topography of PVDF nanocomposite films was investigated using a scanning electron microscope (SEM) and correlative imaging (CPEM, combinate AFM and SEM). A selection of 2D CPEM images was used for a detailed study of the spherulitic morphologies (grains size around 6–10 μm) and surface roughness (value of 50–68 nm). The chemical interactions were evaluated by Fourier transform infrared spectroscopy (FTIR). Dominant polar γ-phase in the original PVDF, PVDF_ZnO and PVDF_ZnO/V, the most stable non-polar α-phase in the PVDF_V_CH nanocomposite film and mixture of γ and α phases in the PVDF_V and PVDF_ZnO/V_CH nanocomposite films were confirmed. Moderately hydrophilic PVDF nanocomposite films with water contact angle values (WCA) in the range of 58°–69° showed surface stability with respect to the Zeta potential values. The effect of positive or negative Zeta-potential values of nanofillers (ζ_n_) on the resulting negative Zeta-potential values (ζ) of PVDF nanocomposite films was demonstrated. Interaction of PVDF chains with hydroxy groups of vermiculite and amino and imino groups of CH caused transformation of γ-phase to α. The friction properties were evaluated based on the wear testing and mechanical properties were evaluated from the tensile tests based on Young’s modulus (E) and tensile strength (Rm) values. Used nanofillers caused decreasing of friction and mechanical properties of PVDF nanocomposite material films.

## 1. Introduction

Polyvinylidene difluoride (PVDF) belongs to frequently used materials in various application areas, including semiconductor equipment components, constructions, fluid systems (oil-and-gas) and food industries. PVDF is mainly used where attention is paid to an excellent chemical resistance, a high degree of purity, excellent mechanical properties and abrasion resistance.

The new functional nanomaterials are PVDF composites (nanocomposites), which have been broadly applied in the fields of membranes and biomedicine industry. Due to their light weight, thinness, and good compatibility, their good strength is required in practical applications [1]. Surface and morphological [2] types of phase transformations [3], mechanical [4] and also friction properties are the important considerations that decide the areas of application and the quality of the PVDF final products [5].

PVDF is a semicrystalline thermoplastic polymer with polar type voids [6]. PVDF properties are dependent on its crystalline structure, which has five common crystalline phases/polymorphs: α, β, γ, δ and ε. The most frequently occurring phases are non-polar α-phase, polar β-phase, and polar γ-phase, which are different in macromolecular chain conformations [2]. The α-phase dominates among PVDF nanocomposite materials, and it is easily obtained because of its thermodynamic stability; the β and the γ phases cannot be shaped naturally. The crystalline phases can be transformed using several preparation methods and mainly by their preparation conditions [7], but also nanofiller additions [8]. Adding nanofillers such as carbon nanotubes [4], titanium oxide [9,10], or silica nanoparticles [11] to promote the polar phases of the PVDF nanocomposite materials has recently been reported.

The type of nanofiller, especially its morphology, surface properties, and agglomeration ability, affect not only the crystalline phase but also the morphology, roughness, surface and mechanical stability of the PVDF nanocomposite materials. The mechanical properties of polymer materials are strongly correlated with the underlying microstructure of PVDF nanomaterials. On the contrary, fillers based on clay minerals such as montmorillonite [12], cloisite [13] and nanoclay [14] also improve the mechanical and tribological properties of PVDF because the clay at the low content may act as the reinforcing element to bore load and thus decrease the plastic deformation. Non-negligible changes in the PVDF material can be expected from a nanofiller based on vermiculite, which due to surface groups and interlayered spacing can influence the hydrophilic or hydrophobic properties and structural stability of the polymer [15]. It can be assumed that hybrid nanofillers in polymer matrices will show synergistic effects of the organic phase and the inorganic component, especially with advantages including the expected excellent mechanical properties, chemical and structural stabilities.

This work deals with the structural, friction and mechanical properties of PVDF nanocomposite films prepared by the solvent casting method with the use of five nanofiller particle types. In the work, a systematic investigation was undertaken on the influence of the organic (chlorhexidine) and inorganic (zinc oxide nanoparticles) components of vermiculite based nanofillers on the crystal phase changes in PVDF nanocomposite films. Their influence on surface changes with respect to changes in the roughness and hydrophilic character of the PVDF materials was observed. The resulting changes were always compared with the original PVDF properties. A great contribution of the work is the discovery that an important parameter of these changes is also knowledge of the Zeta-potential of individual nanofillers, which tells us about the stability of the final PVDF nanocomposite films.

## 2. Materials and Methods

### 2.1. Nanofiller Particles and PVDF Nanocomposite Films Preparation

The five nanofiller particle types were used for PVDF nanocomposite films preparation: (1) the natural Mg-vermiculite particles (V, Grena Co., Veselí nad Lužnicí, Czech Republic) with a structural formula (Si_6.32_Al_1.58_Ti_0.1_) (Mg_4.75_Ca_0.34_Fe_0.91_) O_20_ (OH)_4_ (Ca_0.04_ K_0.38_); (2) zinc oxide nanoparticles prepared by the sonochemical process (ZnO) and nanocomposite particle samples prepared by the sonochemical process; (3) zinc oxide/vermiculite (ZnO/V); (4) vermiculite_chlorhexidine (V_CH); and (5) zinc oxide/vermiculite_chlorhexidine (ZnO/V_CH). Detailed procedures and conditions of the sonochemical process for the preparation of nanocomposite particles were used based on the method proposed in our previous work [16,17]. Table 1 summarizes particles characteristics as a particle size (evaluated based on mode d_m_ diameters values), specific surface area (SSA) and ZnO crystallite size (Lc).

A solvent casting method, with dichloromethane (DCM, Sigma Aldrich, Czech Republic) as solvent, was used for the preparation of PVDF films. The mixture of the 1 g of PVDF pellets (Sigma Aldrich, Czech Republic), 10 mL of *N,N*-dimethylformamide (Sigma Aldrich, Czech Republic, M_w_ = 73.095 g/mol) and 7 mL of acetone (Sigma Aldrich, Czech Republic, M_w_ = 58.081 g/mol) was stirred at 80 °C for 30 min in an ultrasonic bath until the PVDF pellets were completely dissolved. Then the solution was poured into a Petri dish with a diameter of 11 cm and dried in an oven with continuous suction at 160 °C for 24 h. The nature poly(vinylidene fluoride) film was denoted as PVDF.

The PVDF nanocomposite films were prepared under the same conditions as the original PVDF film, but to the PVDF solution the 3 wt% of nanofiller particles were additionally added and the solution was intensively mixed (for better dispersion of nanofillers in polymeric solution) in ultrasound bath for 20 min. The nanocomposite films were denoted as PVDF_V, PVDF_ZnO. PVDF_ZnO/V, PVDF_V_CH and PVDF_ZnO/V_CH.

### 2.2. Characterization Methods

The surface topography of the original PVDF and PVDF nanocomposite films and arrangements of the nanofillers in the PVDF matrix were investigated using a scanning transmission electron microscope (STEM, JEOL JSM-7610F Plus, Tokyo, Japan). The samples were not sputtered, and the SEM images were obtained in a low vacuum using a secondary electron detector (SE, LEI).

The correlative imaging (CPEM) combinate correlative probe (AFM, LiteScope™, NenoVision, Brno, Czech Republic) and electron microscopy (STEM, JEOL JSM-7610F Plus, Tokyo, Japan) were used for the detail characterisation of the surface topography and roughness of the PVDF samples. The “in situ” measurement was carried out in non-contact mode with an 8 μm z-linearized dry scanner. The 2D and 3D images and roughness data were evaluated using the Gwyddion 2.55 software.

The water contact angle (WCA) of PVDF and PVDF nanocomposite films was measured using a three-point technique at 22.5 °C, 995 mba and relative humidity 65%. An amount of 0.1 mL of distilled water was deposited onto the surface of the PVDF nanocomposite films using a micropipette; each drop (0.1 mL) was recorded using a Mitutoyo videocamera (Tokyo, Japan) and its images were evaluated using Pixel Fox program (Germany). The examined WCA are the results of 4 repeated measurements.

Electrokinetic analysis (determination of Zeta-potential, ζ) of the PVDF and PVDF nanocomposite films was accomplished on SurPASS Instrument (Anton Paar, Austria). Samples were studied inside the adjustable gap cell in contact with the electrolyte (0.001 mol dm^3^ KCl) at room temperature. For each measurement a pair of polymer films with the same top layer was fixed on two sample holders (with a cross section of 20 × 10 mm^2^ and gap between 100 µm). All samples were measured 6 times at constant pH (pH = 6.6) with a relative error of 5%. For determination of the ζ the streaming current method was used and the Helmholtze–Smoluchowski equation was applied to calculate ζ values.

The Zeta-potential of the PVDF nanocomposite films (ζ) was compared with the Zeta-potential of the nanofiller particles (ζn), which was measured by a nanoparticle analyser (HORIBA Nanopartica SZ-100, Kyoto, Japan) equipped with a microprocessor unit to directly calculate the ζ_n_ values. A quantity of 0.1 g of each nanofiller sample was mechanically mixed with 25 mL of distilled water, and 0.1 mL of the suspension was introduced into the disposable Zeta potential cell. Each data point is an average of 4 measurements realised at 22.5 °C.

The FTIR spectra of the original PVDF and PVDF nanocomposite films were measured by the ATR (attenuated total reflectance, USA) technique. The samples were laid and pressed with a pressure device on a single-reflection diamond ATR crystal. The FTIR spectra were collected using an FT-IR spectrometer, Nicolet iS50 (ThermoScientific, Waltham, MA, USA), with a DTGS detector on a Smart Orbit ATR accessory. The measurement parameters were as follows: spectral region, 4000–400 cm^−1^; spectral resolution, 4 cm^−1^; 64 scans; and Happ-Genzel apodization.

The friction and wear testing of the original PVDF and PVDF nanocomposite films were accomplished using the mechanical tester UMT Tribolab (Bruker Corporation, Billerica, MA, USA) and optical profilometer Contour GTX (Bruker Corporation, USA) by the ball-on-flat method. The steel balls with a diameter of 10 mm and microhardness of 60 HRC were used. The stroke length of the reciprocating movement was set as 10 mm. The test was carried out with a 2 N loading force and frequency of 5 Hz for 5 min, which meant 1000 passes over the sample surface. The wear track depth (the geometry of the wear) was evaluated using a 5× objective in VSI mode. Tilting of the measurement data was performed by plane fitting, and the measured points were evaluated using the Legacy method.

The analysis of the mechanical properties of the PVDF nanocomposite films was carried out using the MTS Criterion Model 43 static testing machine. The tensile test was carried out in accordance with the recommendations of the PN-EN ISO 527 standard. The samples were subjected to a static tensile test at a speed of 50 mm/min. The results of force measurements were collected with an accuracy of 1N. The measurement of each sample was repeated 4 times.

## 3. Results and Discussion

### 3.1. PVDF Nanocomposite Materials Characterization

The PVDF and PVDF nanocomposite films prepared by the ultrasonication assisted solvent casting method with different nanofiller particles were investigated using a scanning electron microscope (SEM) and are show in Figure 1. Surface morphology and roughness of the PVDF film samples was observed from the top surface of the film samples, i.e., opposite to the surface in contact with the glass substrate (bottom surface).

The original PVDF film was formed by regularly repeating hexagonal and cubic grains with an average size (width and length) of 36.8 μm. White particles were visible on the surface of the spherulitic grains due to the presence of surface impurities (all films were characterized without surface treatment) and pores with an average size of 2.8 μm.

PVDF_V nanocomposite film was formed by spherulitic grains of irregular shapes and sizes in the range of 5.1–10.8 μm. Cavities whose diameter did not exceed 4 μm occurred sporadically between these grains. The vermiculite particles (as a reference filler) were found both in the volume of the PVDF matrix and on the surface of the spherulitic grains and did not exceed an average size of 3.8 μm. Compact spherulitic grains of two sizes, 11.5 μm with a predominant irregular hexagonal shape and 7.5 μm of irregular triangular and hexagonal shape, w characterized in PVDF_ZnO nanocomposite films. The spherulitic grains were close together connected by the edges. There were cavities of non-uniform size in the place of the grain tops. On the surface of PVDF_ZnO nanocomposite films white impurities and scratches caused by handling the polymer film were present. The PVDF_ZnO/V and PVDF_V_CH nanocomposite films consisted of spherulitic grains of two sizes, 13.4 μm and 6.4 μm in the case of PVDF_ZnO/V, and 14.0 μm and 6.9 μm in the case of PVDF_V_CH. There were cavities of different sizes and shapes between the individual grains. The ZnO/V and V_CH nanoparticles were incorporated in the volume of the polymer matrix.

The completely compact (non-porous) PVDF_ZnO/V_CH film was formed by large spherulitic grains with sizes in the range of 18–22.4 μm, at the boundaries of which there were particles of ZnO/V_CH nanofillers with a size of 0.7–1.14 μm. These nanoparticles were predominantly oriented as perpendicular or at an angle of 45° in the polymer matrix.

The topography of PVDF and PVDF nanocomposite films was characterized in detail using correlative imaging (CPEM), combined correlative probe (AFM) and scanning electron microscopy (SEM). In the left part of Figure 1 is the visible location of the AFM tip. To characterize the topography and roughness of the individual spherulitic grains, areas with a low occurrence of voids were selected. Individual 2-dimensional (2D) scans and 3-dimensional (3D) profiles are shown in Figure 2.

From 2D CPEM images it is evident that the spherulitic morphologies of the PVDF grains are significantly influenced by the addition of specific nanofillers into the polymer matrix.

The largest spherulitic grains occurred in the PVDF sample with maximum heights at around 8.2 µm, followed by the PVDF_ZnO/V_CH nanocomposite film with heights of 3.9 µm and PVDF_V_CH with heights of 3.7 µm. The highest height differences correspond with the typical lamellae sheets of the PVDF spherulitic grains that produce outwards from their nucleation centres, where morphology is characterized by jammed circular platelets (diameter < 1 μm) and make up the entire volume of PVDF grains.

It should be mentioned that the original PVDF film was formed by two types of spherulitic grains. The first grains (representing a minor part) were formed by regularly branching lamellae sheets, between which lamellae of the dendritic type also appeared in the amorphous region. The second type of grain that predominated in the bulk of the PVDF film were planar spherulite morphologies. Thanks to the presence of these morphologically different grains it was possible to evaluate the differences in the average value of their roughness. While lamellae grains reached a roughness of 52.9 nm (Ra), grains of planar morphology possessed only 27.0 nm (Ra). In the overall context, the roughness of PVDF then reached 31.9 nm (Ra), Table 2.

The spherulitic grains of the PVDF_ZnO/V_CH nanocomposite film were formed by lamellae morphology with a dendritic crystal on the centre. As can be seen from the image, numerous particles with a size of around 200 nm were randomly distributed on the edges of the spherulitic grains. These particles were the ZnO/V_CH nanofillers on the surface of the PVDF matrix and represented a minor size fraction of these nanofiller.

In contrast to the SEM images, based on the 2D (or 3D) CPEM scan of the PVDF_V_CH nanocomposite film it was found that on the surface of the spherulitic grains there were pores occurred in the polymer matrix with an average size of 6 μm and a depth about 1 μm. These pores emerging irregularly on the surface of the entire PVDF_V_CH sample created the so-called hallo effect in the centre of which there were very fine particles. This structural artefact has not been showed in any of the other polymer films.

The smoothest surfaces of spherulite grains were characterized on the surface of the PVDF_V, PVDF_ZnO and PVDF_ZnO/V nanocomposite films. The grains were hexagonal to spherical in shape, made up of ordered lamellae which are connected by an amorphous region, without voids, cracks and other deformations. Only sporadically, dust particles did occur at the grain boundaries.

Surface roughness values were also characterized using the CPEM scan, which correlate with surface morphology of the PVDF films. These values are significantly affected by many spherulitic grain peaks and voids that appeared in the PCEM images. For this reason, the roughness of the PVDF films was evaluated via the average roughness (Ra, measured by profile/linear analysis) and root mean square roughness (RMS) values (Table 2). All were obtained from the ten measurements at different locations of the each PVDF films. The roughness with the lowest values of 31.9 nm (Ra) and 43.2 nm (RMS) was measured and evaluated for the natural PVDF film. Nanofillers in the PVDF matrix caused an increase in the roughness values in the range of Ra value from 37.2 nm (for PVDF_ZnO/V_CH sample) to 53.9 nm (for PVDF_V sample). The root means square roughness (RMS) reached higher values than the average roughness values (Ra), while the decreasing character was maintained. The lowest roughness was measured for PVDF film (RMS = 43.2 nm), and the highest values were evaluated for PVDF_V (RMS = 68.0 nm) and PVDF_V_CH (RMS = 59.5 nm) nanocomposite films. With regard to roughness values, the ascending character of these values was recorded in the following order for nanocomposite films: PVDF_ZnO/V < PVDF_ZnO/V_CH < PVDF_ZnO < PVDF_V_CH < PVDF_V. It was found that the change in roughness values does not correlate with lamellar or planar surface topography of the PVDF nanocomposite materials but with the size of the specific surface area of the individual nanofillers. It can be stated that the roughness of polymer nanocomposite films decreases with the decreasing value of the specific surface area of nanofillers.

The surface wettability of the PVDF samples was evaluated from the water contact angles (Table 2). The WCA values confirmed the wettable (hydrophilic) surface of all PVDF samples when the lowest WCA value measured for the original PVDF film was 43°. The nanofillers in the PVDF matrix preserved the hydrophilic nature of the nanocomposite films but caused an increase in WCA values in the range of 58°–69°. The WCA values are in good agreement with not only the surface roughness but also surface morphology and topography. The fact that a minor part of the nanofillers in the PVDF_ZnO/V_CH, PVDF_ZnO/V and PVDF_V nanocomposite films was located on the surface of the spherulitic grains had a non-negligible effect on the lower WCA values. It is known that vermiculite particles (V) as part of nanofillers are characterized by a high adsorption capacity and a large surface area (see Table 1) [18], and therefore it can be assumed that they adsorbed water molecules into their interlayer space during the measurement. The lower hydrophilicity of PVDF nanocomposite materials also contributes to the numerous presences of voids located between individual PVDF spherulitic grains. Overall, PVDF nanocomposite films are moderately hydrophilic.

Table 3 presents Zeta-potential values for individual samples. The left column presents values for nanofiller particles (ζ_n_) and the right column values for PVDF nanocomposite films with these nanofillers. The Zeta-potentials of nanoparticles ζ_n_ vary significantly, especially when nanoparticles are covered with chlorohexidine (CH). It changes to the positive values due to a presence of amino and imino groups in chlorohexidine molecules which causes the positive surface charge [19]. The values for PVDF nanocomposite films show the similar trend. The original PVDF film is negatively charged due to presence of fluorine in its structure. Zeta-potential of PVDF nanocomposite films with individual nanoparticles changes according to the Zeta-potential of used nanofillers. When nanofillers are covered with chlorohexidine, Zeta-potential of PVDF changed significantly to the lower negative value (in comparison with original PVDF) due to the presence of amino and imino groups. This change is more significant for ZnO/V_CH in comparison with only V_CH the same as the ZnO/V_CH particles have much positive Zeta-potential. Due to less negative (for ZnO and ZnO/V) or even positive (for V_CH and ZnO/V_CH) surface charge of used nanoparticles it is expected to create strong electrostatic interactions between used nanoparticles and PVDF with a strongly negative surface charge. Due to this it is expected for preparation of stable PVDF nanocomposite films. It is confirmed by very small standard deviation of Zeta-potential during measurement of individual samples (samples are wetted by electrolyte under pressure).

### 3.2. Fourier Transform Infrared Spectroscopy

The FTIR technique is a very useful tool providing us with information about the structure of semi-crystalline polymer PVDF which exists in three basic distinct polymorphs and allows us to distinguish between them [20,21]. The FTIR spectrum of the original PVDF film (Figure 3 and Figure 4) shows typical bands for γ-phase at 1429, 1231 and 835 and 510 cm^−1^ attributed to CH_2_ bending, C-F out-of-plane deformation and CH_2_ rocking vibrations, respectively [21]. In the case of spectra of PVDF_ZnO and PVDF_ZnO/V nanocomposite film (Figure 4) we can observe almost the same vibrations as for original PVDF film which indicates that both nanofillers did not change PVDF polymorph type.

A different situation occurs in the case of FTIR spectrum for PVDF_V nanocomposite film (Figure 3). Except predominant γ-phase in this sample the addition of V nanofiller led to the formation of small amount of α-phase for which following vibrations are typical: 1211, 976, 797, 763, 614 and 531 cm^−1^ which belong to CH_2_ bending, C-H out-of-plane deformation, CH_2_ rocking, CF_2_ bending and sceletal bending, respectively [21].

The presence of organic component chlorhexidine CH in nanofillers caused formation of α-phase, when FTIR spectrum for PVDF_V_CH (Figure 3) confirmed creation solely of this polymorph, on the other hand on combination with ZnO in sample PVDF_ZnO/V_CH FTIR spectrum (Figure 4) showed a mixture of α and γ phases.

It can be assumed that hydroxy groups of vermiculite, as well as amino and imino groups of chlorhexidine through negatively charged PVDF due to the presence of fluorine, cause transformation of PVDF phases from polar γ to non-polar α, but on the contrary, the presence of ZnO inhibits this process. These findings agree with values of Zeta-potentials ζ_n_ for above mentioned nanofillers (Table 3).

### 3.3. Friction and Mechanical Properties of the PVDF Nanocomposite Films

The friction/tribological properties of the PVDF samples were evaluated by static wear tests performed against steel balls by the ball-on-flat method. The testing conditions (1N for 5 min.) were chosen with regard to the simulation of normal intensively loading and wear of the polymer film. Table 4 summarizes the friction coefficients (COF, average values measured during the test based on four repetitions of each test, with the minimum standard deviations ±0.05) and abrasion depths (AD, standard deviations ±0.3 µm). The representative friction/tribological plots are shown in Figure 5 and profilometry images are shown in Figure 6 of PVDF and PVDF nanocomposite films.

From the friction/tribological plots (Figure 5) it is evident that each of the PVDF samples achieves frictional stability (a steady value and smooth polymer surface) after different times of the friction load. While in the case of the PVDF_ZnO/V_CH and PVDF_V_CH nanocomposite films stability is achieved after 15 s, for the PVDF_ZnO and PVDF_V samples it is 25 s. The tribological plots of the original PVDF and PVDF_ZnO/V nanocomposite film continuously increased up to 60 s PVDF samples and reached a steady COF value after 70 s. The COF values measured at the end of the measuring time are shown in Table 4.

The COF values of the PVDF nanocomposite films were in the range (in terms of average values) of 0.79 to 1.07, compared to the original PVDF film with value 0.91. Only the PVDF_V sample showed an increase in the COF value, while the COF value decreased slightly for the other PVDF nanocomposite films. It can be assumed that the vermiculite particles (V) in the PVDF matrix, due to their horizontal orientation, contribute to the increase in COF values. Similar changes in COF values were also observed in [22]. However, the values of abrasion depths (AD) and abrasion track width (ATW), which are shown in Table 4, have more description about resistance of the nanocomposite material against frictional action.

The smallest penetration of the steel ball into the polymer materials and therefore the highest resistance of the material against damage/abrasion occurred in the PVDF and PVDF_ZnO samples, when the values of abrasion depths (AD) reached the lowest values of 4.6 and 6.1 µm. On the contrary, the largest penetration was measured for the PVDF_V_CH nanocomposite material, namely 20.7 µm (AD). Other values of abrasion depths (AD) ranged from 12.1 to 16.6 µm. The abrasion track width (ATW) values also correspond to these values. The AD and ATW values indicate that the PVDF and PVDF_ZnO nanocomposite plates exhibited better wear resistance.

Figure 6 shows the representative optical images of the polymer material surfaces after friction/tribological tests, and the points that were used to evaluate the abrasion depths values (AD) are also marked. It can be seen that in all polymer samples a smooth sliding plane was formed at the point of contact and frictional movement of the steel ball. The PVDF and PVDF_ZnO nanocomposite films showed visible migration of the PVDF matrix and the formation of a polymer edge. This is most evident in the PVDF_ZnO nanocomposite film, where polymer craters with an average size of 10 μm are visible on the edges. In the case of the PVDF sample, the formation of structural defects in the form of holes are visible (blue area visible in the image), which were also visible to a lesser extent in the case of the PVDF_ZnO/V nanocomposite film.

The most visible change in the structure after the friction test was for the PVDF_ZnO/V_CH nanocomposite film. The originally compact structure without visible cavities was formed by parts of mutually separated spherulitic grains, resulting in the most porous material of all the studied samples. The PVDF_ZnO/V_CH nanocomposite film shows the lowest Zeta-potential values (−16.6 mV) of all samples, which is a prerequisite for the least stable nanocomposite film. At the same time, as the only nanocomposite film, it had visible fractions of nanofillers occurring at the boundaries of the spherulitic grains. It can be assumed that the acting frictional movement is so intensive that it causes disruption of the bonds between the spherulitic grains and therefore to the total destruction and structural changes in the PVDF_ZnO/V_CH nanocomposite film. However, it is evident that during the friction test a smooth sliding plane occurs at the point of contact of the steel ball with the surface, as is the case with other PVDF nanocomposite films. This fact is confirmed by average AD and ATW friction values in PVDF_ZnO/V_CH nanocomposite film.

The least friction-resistant material was the PVDF_V_CH nanocomposite film, which reached the highest AD and ATW values. Here it can be assumed that the V_CH nanofiller created a friction layer that allowed the steel ball to move smoothly over the tested polymer surface without structurally disrupting the PVDF_V_CH nanocomposite film. Similar behavior can be assumed for the PVDF_V nanocomposite material, however, there is not such a deep penetration of the steel ball into the material, which is declared by the AD and ATW values and at the same time the highest COF value.

The mechanical properties of the PVDF nanocomposite films were evaluated based on the tensile tests. The results of the tests carried out are presented in Table 5 and the tensile test curves are shown in Figure 7. On the basis of the obtained test results, the following parameters were calculated: Young’s modulus (E), tensile strength (Rm), maximum force (Fmax) and maximum strain (Smax).

When analysing the obtained test results, it was noticed that the highest tensile strength values were obtained for PVDF (Rm = 28 MPa) [23] and the lowest for PVDF_ZnO/V_CH (6 MPa) nanocomposite films. In the case of PVDF_V, PVDF_ZnO, PVDF_ZnO/V, PVDF_V_CH nanocomposite films, the obtained values were similar and ranged in value 13–19 MPa. The PVDF_V nanocomposite film was characterized by the highest stiffness (E = 568 MPa), and thus the greatest Young modulus, while the greatest flexibility was demonstrated by the PVDF_ZnO/V and PVDF_ZnO/V_CH nanocomposite films with the 213 MPa, 214 MPa, respectively.

From the tensile stress–strain curves of the PVDF and PVDF nanocomposite films (Figure 7) it is evident that the most susceptible to deformation turned out PVDF_ZnO/V nanocomposite film with the Smax = 0.985 mm.mm^−1^ compared with original PVDF [23], which was characterized by the lowest deformation Smax = 0.065 mm.mm^−1^. In the case of other materials, the obtained differences were not so significant. As for the breaking strength (Fmax), the highest value was recorded for PVDF_V (52 N), while the lowest for PVDF_ZnO/V_CH (8.5 N). The original PVDF and PVDF_V_CH nanocomposite film had the same breaking strength value of 41 N.

The mechanical test values of the PVDF nanocomposite films were compared with mechanical test results obtained for the PVDF nanofibers material with identical nanofillers [24]. The decrease in the Young’s modulus (E) was observed for the PVDF, PVDF_ZnO, PVDF_ZnO/V and PVDF_ZnO/V_CH nanocomposite films. The percentage decrease in these values was determined, which for individual materials was as follows: PVDF (−22%), PVDF_ZnO (−27%), PVDF_ZnO/V (−43%), PVDF_ZnO/V_CH (−31%). Only in the case of the PVDF_V_CH nanocomposite film was an increase in the value of Young’s modulus noted, by 14%. In the case of the maximum breaking stress Rm, a decrease of an average of 68% was recorded for all materials. The largest decrease (Rm) was recorded for the PVDF_ZnO/V_CH nanocomposite film, which was 88%. The obtained research results were compared with the results of other authors [23,25]. The value of deformations and maximum stresses was obtained at a similar level.

## 4. Conclusions

The five types of nanofiller particle were used for PVDF nanocomposite films preparation via the solvent casting method. The effect of nanofillers in PVDF nanocomposite films on the structural, phase, friction and mechanical properties was examined and compared with that of the original PVDF film.

The SEM images showed that the V, ZnO and ZnO/V nanofillers with the highest Zeta-potential values (ζ) of the PVDF nanocomposite films created the spherulitic morphology with the grains size around 6–10 μm. The smoothest surfaces of the PVDF spherulite grains and average values of surface roughness in the interval value of 50–68 nm (RMS) were evaluated from the CPEM scan. Nanofillers with the organic component CH (V_CH, ZnO/V_CH) contribute to the lamellar growth of spherulitic grains similarly to the original PVDF film. These nanofillers reduce, due to their positive Zeta-potential ζn, the Zeta-potential (ζ) of the PVDF nanocomposite films surfaces to the lowest values, which is also a consequence of the occurrence nanofillers at the boundaries of spherulitic grains (PVDF_ZnO/V_CH sample) and pores (PVDF_V_CH sample). This physical property of the two mentioned nanofillers connected to amino and imino groups of CH was also related to their contribution to the transformation of the PVDF phases from polar γ to non-polar α confirmed by FTIR. On the other hand, it was found that ZnO inhibits this process.

The nanofillers had no significant effect on the average values of the friction coefficients (COF = 0.79–1.07). PVDF_V_CH nanocomposite film, which reached the highest AD and ATW values, was evaluated as the least resistant material to frictional wear/abrasion. On the other hand, the PVDF_ZnO nanocomposite film is the most resistant to the penetration with the tested steel ball into the PVDF matrix as shown by the optical profilometry images. From the tensile tests it is evident that the nanofillers caused a decrease in the tensile strength values (Rm) and stiffness (E). Especially for nanofillers containing ZnO nanoparticles, the lowest Rm values were measured in the range of 213–284 MPa. On the contrary, V and V_CH nanofillers significantly increased the stiffness (and respectively brittleness) of the PVDF nanocomposite films.

Overall, it was found that the used nanofillers caused a refinement of the initially large spherulitic grains, while their size was preserved in the case of PVDF_ZnO/V_CH. With the refinement of the structure, there was an increase in the average values of the surface roughness. Nanofillers caused a decrease in friction coefficients (COF) and Young’s modulus (E). The exceptions were V and V_CH nanofillers, which increased these values. These results provide an alternative route for the preparation of new PVDF nanocomposite films using cheap and technologically simply prepared nanofillers that form particles of natural clay minerals. They enable the homogeneous incorporation of both organic and inorganic components into the polymer matrix.

## Figures and Tables

**Figure 1 polymers-14-03831-f001:**
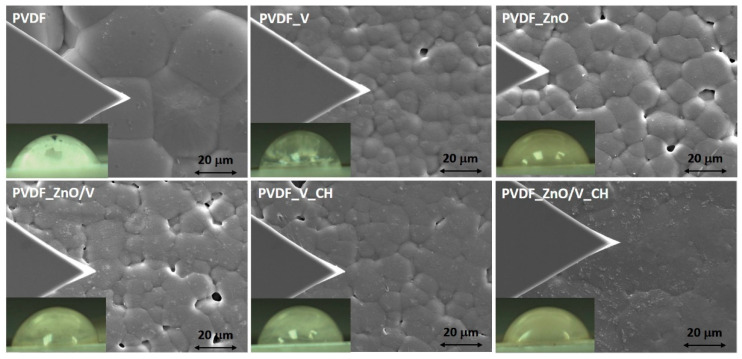
SEM images of the PVDF and PVDF nanocomposite films and their optical images of the water drops. The triangle on the left side of the images represents the AFM tip.

**Figure 2 polymers-14-03831-f002:**
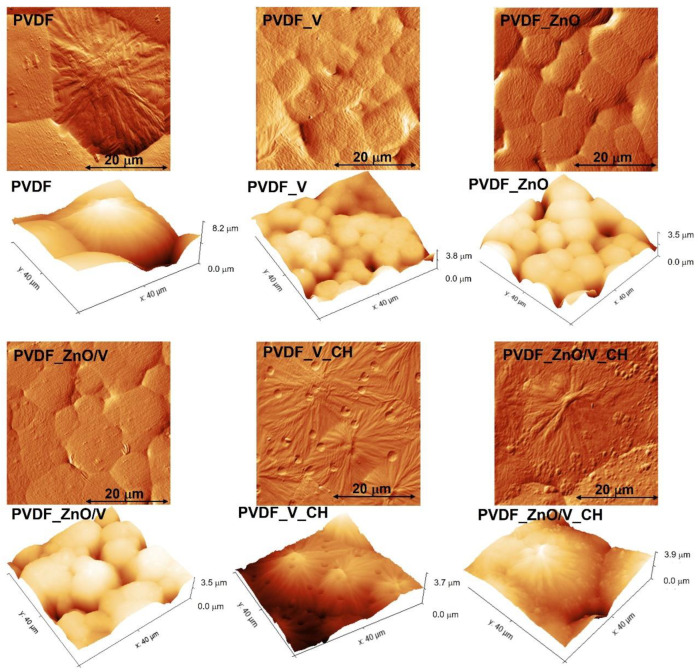
2D and 3D CPEM images of the PVDF and PVDF nanocomposite surfaces. Scanning area: 40 μm × 40 μm.

**Figure 3 polymers-14-03831-f003:**
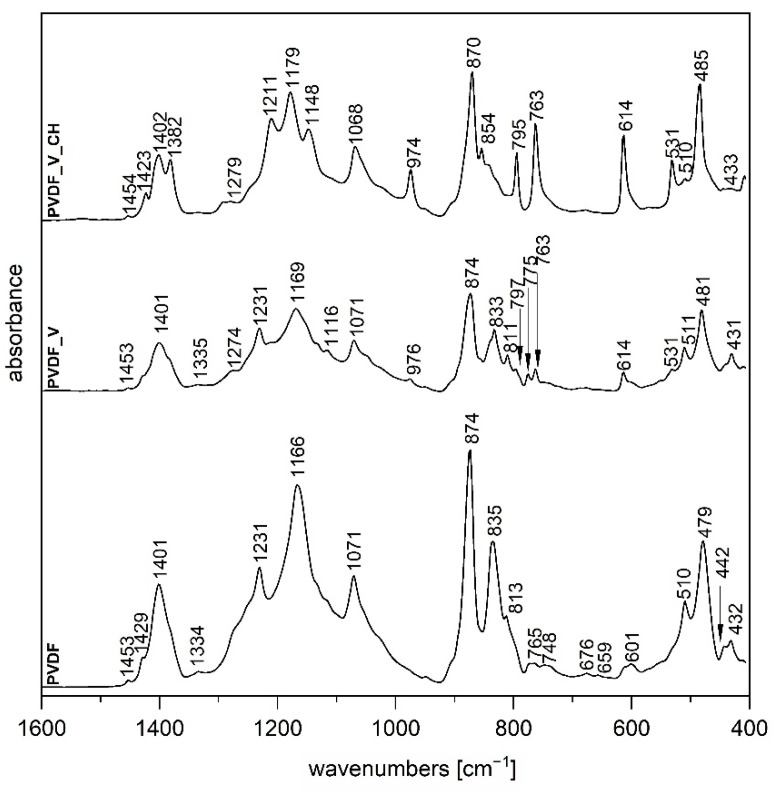
FTIR spectra of PVDF, PVDF_V and PVDF_V_CH nanocomposite films.

**Figure 4 polymers-14-03831-f004:**
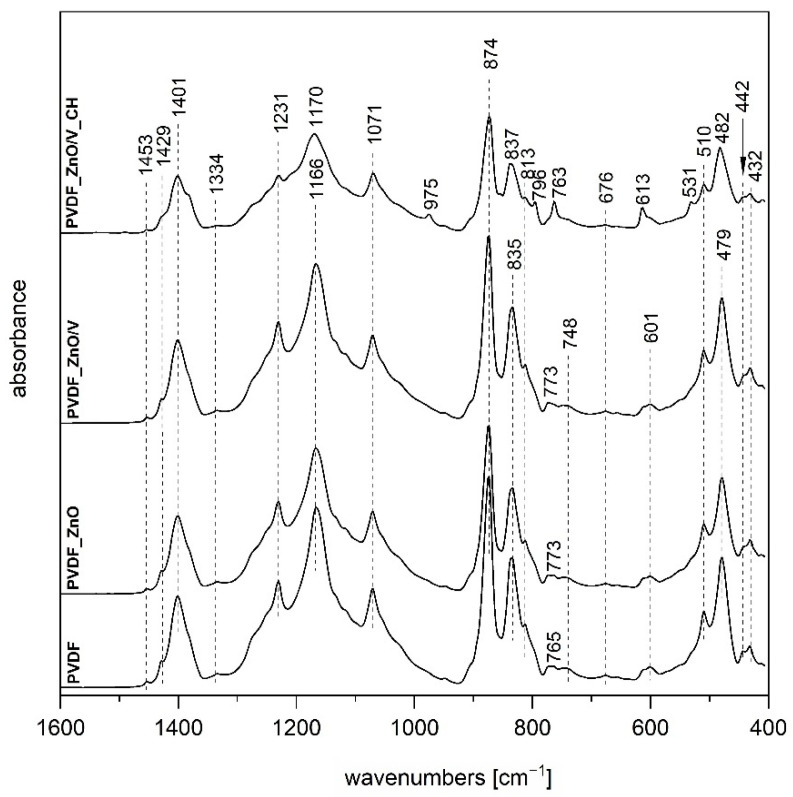
FTIR spectra of PVDF, PVDF_ZnO, PVDF_ZnO/V and PVDF_ZnO/V_CH nanocomposite films.

**Figure 5 polymers-14-03831-f005:**
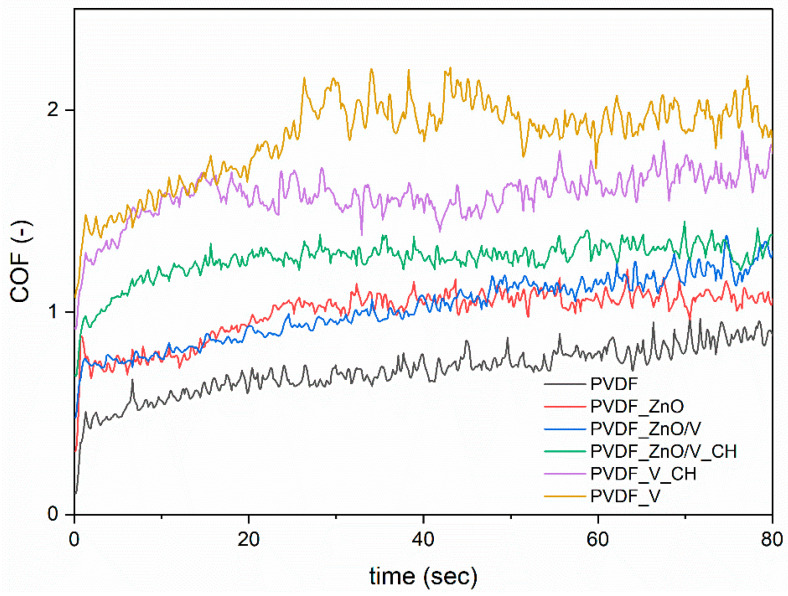
The first part of tribological plots of PVDF and PVDF nanocomposite films.

**Figure 6 polymers-14-03831-f006:**
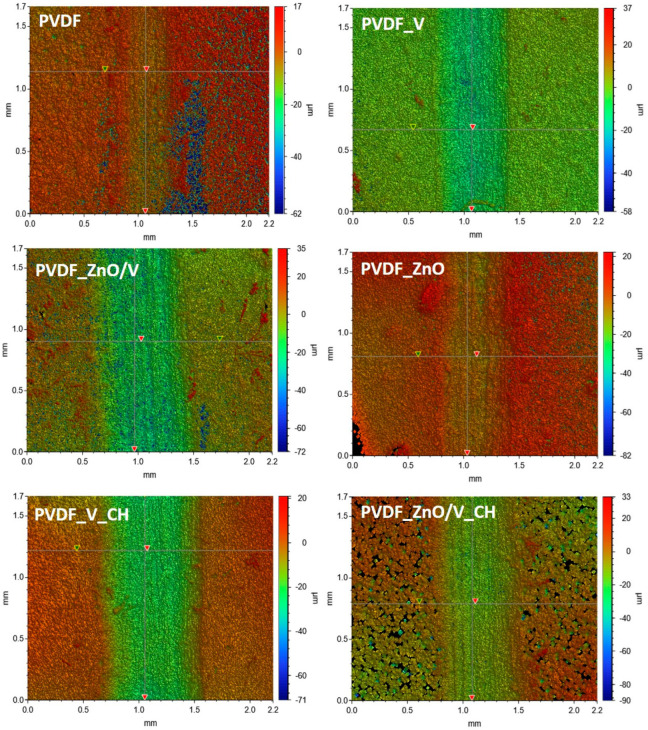
The optical profilometry images against a steel ball after friction/tribological test of the original PVDF and PVDF_V, PVDF_ZnO. PVDF_ZnO/V, PVDF_V_CH and PVDF_ZnO/V_CH nanocomposite films.

**Figure 7 polymers-14-03831-f007:**
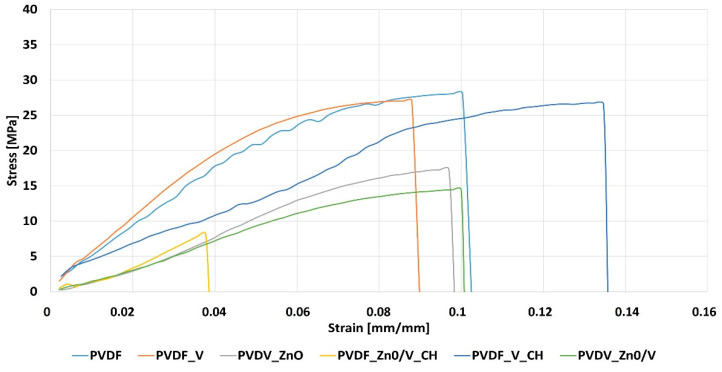
The tensile stress–strain curves of the PVDF and PVDF nanocomposite materials.

**Table 1 polymers-14-03831-t001:** The nanofiller particles characteristics: mode particle size (d_m_), specific surface area (SSA) and ZnO crystallite size (Lc).

Nanofiller Samples	d_m_ (µm)	SSA (m^2^·g^−1^)	Lc (nm)
V	12.4	90.6	-
ZnO	0.15 2.98	52.7	16.0
ZnO/V	0.17 10.10	23.6	12.36
V_CH	0.29 11.5	32.0	-
ZnO/V_CH	0.23 11.6	24.3	7.05

**Table 2 polymers-14-03831-t002:** The average values of the surface roughness (Ra, RMS) evaluated from the AFM measurements of the PVDF surfaces. Water contact angles values (WCA). Standard deviations are provided in parentheses.

PVDF Nanocomposite Films	Ra (nm)	RMS (nm)	WCA (°)
PVDF	31.9 ± 11.9	43.2 ± 15.3	43 ± 0.2
PVDF_V	53.9 ± 17.1	68.0 ± 22.0	64 ± 4.2
PVDF_ZnO	44.4 ± 15.4	59.0 ± 19.6	69 ± 4.4
PVDF_ZnO/V	38.1 ± 9.8	50.2 ± 14.7	64 ± 3.9
PVDF_V_CH	45.0 ± 9.2	59.5 ± 14.3	67 ± 3.0
PVDF_ZnO/V_CH	37.2 ± 7.0	52.4 ± 13.3	58 ± 5.2

**Table 3 polymers-14-03831-t003:** Zeta-potential (ζ_n_) of the nanofillers PVDF and Zeta-potential (ζ) of the PVDF nanocomposite films with standard deviations (S.D.) at pH = 6.6.

Nanofillers	ζ_n_ (mV)	PVDF and PVDF Nanocomposite Films	ζ (mV)
-	-	PVDF	−63.5 ± 1.4
V	−60.0 ± 3.5	PVDF_V	−60.1 ± 1.5
ZnO	−39.8 ± 1.1	PVDF_ZnO	−58.8 ± 0.6
ZnO/V	−20.6 ± 0.8	PVDF_ZnO/V	−53.7 ± 0.0
V_CH	+19.8 ± 1.6	PVDF_V_CH	−36.1 ± 0.2
ZnO/V_CH	+36.7 ± 2.2	PVDF_ZnO/V_CH	−16.6 ± 0.3

**Table 4 polymers-14-03831-t004:** The friction coefficients (COF), abrasion depths (AD) and abrasion track width (ATW) of the PVDF and PVDF nanocomposite films.

PVDF and PVDF Nanocomposite Films	COF (-)	AD (µm)	ATW (µm)
PVDF	0.91	4.6	382
PVDF_V	1.07	12.1	707
PVDF_ZnO	0.86	6.1	660
PVDF_ZnO/V	0.83	16.6	897
PVDF_V_CH	0.87	20.7	981
PVDF_ZnO/V_CH	0.79	13.0	783

**Table 5 polymers-14-03831-t005:** Test results with standard deviations of the PVDF and PVDF nanocomposite films.

PVDF and PVDF Nanocomposite Films	E (MPa)	Rm (MPa)	Fmax (N)	Smax (mm·mm^−1^)
PVDF	320	28	41	0.07
PVDF_V	568	14	52	0.10
PVDF_ZnO	284	19	38	0.19
PVDF_ZnO/V	213	13	45	0.99
PVDF_ZnO/V_CH	214	6	8.5	0.41
PVDF_V_CH	428	13	41	0.12

## Data Availability

The data used to support the findings of this study are available from the corresponding author upon request.

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
