# Peer review of "Hybrid Nanofillers Creating the Stable PVDF Nanocomposite Films and Their Effect on the Friction and Mechanical Properties"

_polymers, 2022, doi:10.3390/polym14183831_

Round 1

Reviewer 1 Report

Re-write objectives part in the end of introduction.

Many sentences do not have the correct punctuation and it is difficult to read the text.

English should be improved; grammar need for enhancement in many sentences and paragraphs.

Please include some latest research findings, updated reviews in introduction and discussion part related to the topic during 2021-2022.

Please check the References in-text and end-list for uniformity in MDPI style.

Reviewer 2 Report

In this manuscript, the authors studied the PVDF nanocomposite films with hybrid nanofillers. They carried out various methods to investigate the properties of the films, including structural, phase, friction and mechanical properties. The study having abundant data is solid, and of important in the area of composite materials. Therefore, I think the work is well-suitable for publishing on Polymers. I have only minor questions and suggestions, which I hope could help to further polish the manuscript.

1. The abstract is quite long and can be further simplified. Besides describing what has been measured, it is meaningful to summarize the main findings in the abstract also.

2. Please reorganize the keywords, some of them are too special, e.g., “Zeta - potential of nanofillers and PVDF films”, and some of them are too general.

3. Figure 5, the friction coefficient of PVDF_V is the highest. The authors assumed that the vermiculite particles (V) serves as a lubricant. Normally a lubricant helps to decrease the friction coefficient, in contrast to the speculation here.

4. Please be careful to the author contributions. I find it is unsuitable that two unrelated people (M.H. and M.B.) contributes to the work are not on the author list, their full names are even not shown. Meanwhile, one on the author list didn’t do anything (K. J.), and one is in acknowledgement but not author contributions (L.P.).  A reasonable clarification would be very helpful for protecting this high standard work.

5. For the conclusions, I also suggest a brief and clean way to reorganize that. It is currently difficult to see, for a certain kind of hybrid film, what property is improved and what is the advantage. I got more or less lost in these paragraphs because the five materials are interweaved without clear order.

Reviewer 4 Report

Comments:

a.     Fig. 1 shows six SEM graphs. All these 6 micrographs are very poorly resolved and hardly can be evidence. Re-characterization to produce better results is necessary. In addition, the scale bars (magnifications) on graphs are invisible. With such obscured graphs, the discussion texts in 3.1 paragraph do not carry any physical meanings.

b.     This referee cannot see the magnifications in Fig. 1 at all. But this referee supposed that the magnification is way too low and not high enough to discuss about the morphology details of nano-composites. Not only they should re-characterize SEM, but also they should use properly high magnifications.

c.      Same comments apply to the AFM microscopy graphs in Fig. 2. Those 3D images (insets) between 2D graphs are obscured, and poorly presented. Although the AFM 2D phase images are barely OK, the magnification is too low to show any ZnO particles, distribution, etc.

d.     Curves in Fig 5.  Authors stated: “The representative friction/tribological plots are shown in Fig 5 and profilometry images are shown in Fig. 6 of PVDF and PVDF nanocomposite films.”  Other than telling readers that “the friction/tribological plots are shown in Fig 5”, which readers know for sure form the figure captions, but authors did not discuss or analyze anything about the physical meanings of these numerous friction/tribological curves in Fig. 5. 

e.     Furthermore, it is very confusing to readers that along with Fig. 5, authors stated: “profilometry images are shown in Fig. 6 of PVDF and PVDF nanocomposite films.”  Why mix Fig. 5 with Fig. 6? And again, readers know for sure that profilometry images are shown in Fig. 6 of PVDF and PVDF nanocomposite films, but what else physical meanings are to be discussed in Fig. 6?

f.       Fig. 7. Stress-strain curves for specimens (PVDF and its composites) are shown. What are the respective modulus for these samples? Furthermore, why the modulus of PVDF (magenta color) is greater than those of PVDF-ZnO (or other particles, V, V_CH, etc.) nano-composites (yellow, green, and gray colors)? Adding ZnO (or V, V_CH) fillers into PDVF leads to a reduction of modulus? Why?

g.     Caption of “3.2.Fouriertransforminfraredspectroscopy.” needs to be fixed.

h.     Figure 6 is for “The optical profilometry images against a steel ball..”.  First of all, the marks on graphs are almost invisible to readers. Secondly, what is the meaning of the different color-stripes in the graphs?  Thirdly, the scale indications in graphs are again not visible to readers!

i.       Clarity and brevity of both Conclusion and Abstract should be improved. The work evaluated five types of nanofiller particles used for PVDF nanocomposite films. Exactly, which types give the best performance, and which others are no good, and why? And what do nano-composites perform with respect to neat PVDF?

j.       It appears Conclusion states about one portion of data and results, while Abstract states another different portion of data/results. Overall, it gives impression that the Conclusion and Abstract are not correlated with each other to give a major aim of this work? Writing of these two critical texts of manuscript should be carefully improved.

Reviewer 5 Report

Referee Report

on paper “ Hybrid nanofillers creating the stable PVDF nanocomposite films and their effect on the friction and mechanical properties “ (polymers-1903025) by authors Karla ÄŒech Barabaszová, Sylva Holešová, Lukáš Plesník, Zdeňka Kolská, Kamil Joszko and Bożena Gzik-Zroska submitted to Polymers

This is interesting review paper. It reports the preparation and investigation of the structure and mechanical properties of the polyvinylidene difluoride (PVDF) nanocomposite films synthesized via the solvent casting method using the Vermiculite (V), zinc oxide nanoparticles (ZnO), zinc oxide/vermiculite (ZnO/V), zinc oxide/vermiculite-chlorhexidine (ZnO/V_CH) and vermiculite-chlorhexidine (V_CH) as nanofillers. Dominant polar g-phase in the original PVDF, PVDF_ZnO and PVDF_ZnO/V, the most stable non-polar a-phase in the PVDF_V_CH nanocomposite film and mixture of g and a phases in the PVDF_V and VDF_ZnO/V_CH nanocomposite films were confirmed. Moderately hydrophilic PVDF nanocomposite films with water contact angle values (WCA) in the range of 58 - 69°showed surface stability with respect to the Zeta potential values. The effect of positive or negative Zeta - potential values of nanofillers (zn) on the resulting negative Zeta - potential values (z) of PVDF nanocomposite films was demonstrated. The presented results are reliable without any doubts. However, I have some comments and additions. I would like to note a few points to improve the paper before it can be published:

1.    The authors should mention in 1. Introduction some interesting information about nanocomposite materials are perspective for practical applications:

(1). M.A. Almessiere, A.V. Trukhanov, Y. Slimani, K.Y. You, S.V. Trukhanov, E.L. Trukhanova, F. Esa, A. Sadaqat, K. Chaudhary, M. Zdorovets, A. Baykal, Correlation between composition and electrodynamics properties in nanocomposites based on hard/soft ferrimagnetics with strong exchange coupling, Nanomaterials 9 (2019) 202. https://doi.org/10.3390/nano9020202.

2.    The authors should mention in 1. Introduction such experimental methods of non-destructive testing and determination of microstresses in materials as X-ray or/and neutron diffraction:

(2). S.V. Trukhanov, A.V. Trukhanov, V.A. Turchenko, V.G. Kostishyn, L.V. Panina, I.S. Kazakevich, A.M. Balagurov, Structure and magnetic properties of BaFe11.9In0.1O19 hexaferrite in a wide temperature range, J. Alloys Compd. 689 (2016) 383-393. https://doi.org/10.1016/j.jallcom.2016.07.309.

3.    The authors should mention in 1. Introduction some experimental methods for assessing surface tension, friction and wear in materials:

(3). M.A. Darwish, T.I. Zubar, O.D. Kanafyev, D. Zhou, E.L. Trukhanova, S.V. Trukhanov, A.V. Trukhanov, A.M. Henaish, Combined effect of microstructure, surface energy, and adhesion force on the friction of PVA/ferrite spinel nanocomposites, Nanomaterals 12 (2022) 1998. https://doi.org/10.3390/nano12121998.

4.    The proposed 3 papers should be inserted in References.

The paper should be sent to me for the second analysis after the moderate revisions.

Round 2

Reviewer 4 Report

This is revised version V2. Authors have done revisions to respond to most of comments, though not all. e.g. , they did not address comments-d and e, other than saying they kept the original discussion texts....

No further comments.

Minor spell checks are needed.